# LEARNING MULTI-MODAL REPRESENTATIONS UNDER INCOMPLETE DATA VIA DUAL LEVEL ALIGNMENTS

## ABSTRACT

Our goal is to learn modality-free representations of a wide variety of entity types (e.g., text, image, object), that can be applied to multi-modal tasks under incomplete data (e.g., noisy data or missing modality information). While conventional methods train models over modality-specific features, (e.g., image features via visual encoding), and decode them into their contextual representations of specific modalities (e.g., images and text), our framework, *Multiple2Vec* (Mul2vec), is based on the idea that these features and the corresponding text are different views of the same entity, and learns semantic representations without directly using modality-specific features. Mul2vec is a framework consisting of NTF, and training objectives, DLM and ILM. Since this idea implies that similar entities have similar representations even on a dual-level (contextual and semantic), Mul2vec aligns them and optimizes the semantic representations with the corresponding contextual representations. Experiments show that Mul2vec learns semantic representations, and contributes to pre-trained models for downstream tasks under incomplete data.

## 1 INTRODUCTION

Recently, Transformer-based architectures Singh et al. (2022); Hu & Singh (2021); Radford et al. (2021), and Variational AutoEncoder (VAE) Kingma & Welling (2014) based architectures have been applied to multi-modal tasks using modality-specific features, and are tackling the modality gap problem to improve task performance Poklukar et al. (2022); Duan et al. (2022).

Our challenge is to learn the modality-free representations that subsume modality-specific features and texts, while facing missing modality information or noisy data, and mitigating modality collapse. We refer to both text and modality-specific features as **local information**, and the representations obtained from these features as **contextual representations**, as shown in Figure 1(left). Our approach is based on the idea that modality-specific features and the corresponding text share underlying same information, **global information**, as features and text are different views of the same entity. This idea motivates us to treat texts as modality-independent representations, and propose a learning framework, *Multiple2Vec* (Mul2vec). To enable Mul2vec to learn under incomplete data, and mitigate the collapse problem, we A1) extend the tensor factorization to NTF, which can cooperate with pre-trained multi-modal models or the text encoders, and A2) propose objectives that learn modality-free representations in the multi-task learning process. Unlike other models, Mul2vec uses latent variables to quantify global information as **topics**, where these topics are discovered from context-independent information, word-occurrence, and sharing between entities. As topics can align various modal type entities through the aspect of semantics, Mul2vec learns **semantic representations** from topics. That is, the novelty of Mul2vec lies in 1) focusing the semantic representations as modality-free representations, and 2) aligning different modal-type entities in dual-level (contextual and semantic) representations rather than contextual representations.

Experiments support our idea, and show the advantage of Mul2vec that it can compensate for the incomplete features of each other, be compatible with various pre-trained models, and strengthen them, as shown in **4.2**, and contribute to multi-modal tasks by providing robust representations while avoiding the modality gap and tackling both incomplete and noisy data, as shown in **4.3**.

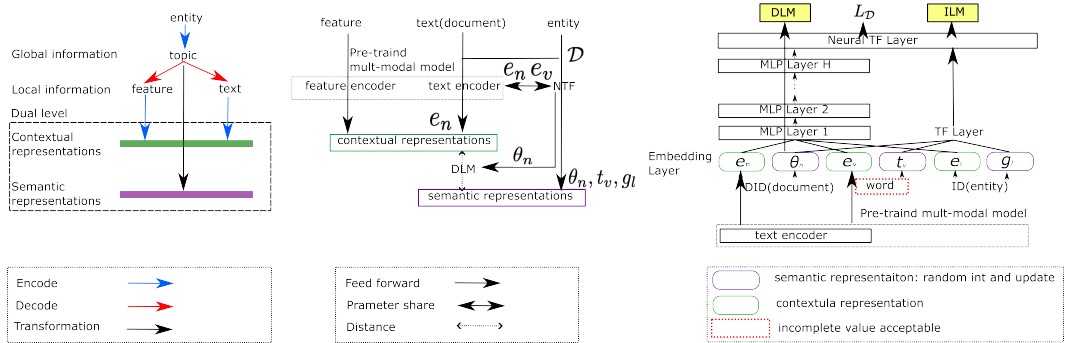

Figure 1: (left) The difference between global/local information and dual-level (contextual and semantic) representations, (center) Information pipeline between feature encoder, text encoder, and NTF, DLM: Given input data $\mathcal{D}$, NTF shares contextual representations with the text encoder, and updates semantic representations, where the contextual representations of $n$-th document, $l$-th entity ID, $v$-th word, and their semantic representations correspond $e_n, e_l, e_v$, and $\theta_n, g_l, t_v$, respectively. and (right) NTF with $L_\mathcal{D}$, DLM and ILM: NTF consists of Tensor Factorization and MLP, where DID, and ID are the identifier of document, and entity, respectively. NTF receives $e_n$ and $e_v$ from the text encoder and fixes them through its training, while updating other representations.

## 2 RELATED WORK

Cross or multi-modal retrieval Balaneshin-kordan & Kotov (2018); Carvalho et al. (2018) aims to find objects in response to textual queries. Conventional methods learn representations using modality-specific features such as RGB values Mithun et al. (2019) and raw waveforms Tseng et al. (2021) for video and acoustic data, respectively. While V+L models Lu et al. (2019); Tan & Bansal (2019); Li et al. (2020a); Su et al. (2020); Chen et al. (2020) use combined visual-linguistic input, or are designed for individual modalities, they often rely on domain-specific assumptions Jaegle et al. (2021), and are developed on a single modality Baevski et al. (2022). CLIP Radford et al. (2021) and ALIGN Jia et al. (2021) require large training resources and massive amounts of data to achieve good alignment. COTS Lu et al. (2022) enhances cross-modal interaction by exploiting token-level, and task-level interaction via momentum contrastive learning. EI-CLIP Ma et al. (2022) aims to alleviate the limitations of CLIP using the theory of causal intervention. While these models learn contextual representations and optimize their similarities using these representations, Mul2vec learns semantic representations and optimizes their similarities on the dual-level representations.

To learn modality-free representations, Mul2vec focuses on the text (document) associated with each entity as a common feature, the modality-independent feature. As shown in Figure 1(center), Mul2vec differs from the state-of-the-art Transformer-based architecture Singh et al. (2022); Hu & Singh (2021); Radford et al. (2021) in that it 1) learns the dual-level (contextual and semantic) representations rather than just the contextual level, 2) performs compensation rather than reconstruction, 3) and derives both the framework and tasks based on this direction. Mul2vec does not depend on Transformer based models using an attention bottleneck Jaegle et al. (2021), masked autoencoder He et al. (2022) modality-specific masking strategies Baevski et al. (2022); Xie et al. (2022), teacher/student training task Baevski et al. (2022), masked prediction Baevski et al. (2022), and self-distillation without labels Caron et al. (2021) as additional tasks. As for training tasks, ID Linguistic Matching (ILM) can jointly pre-train models towards a specific modality, e.g., text and images, together, like other frameworks Akbari et al. (2021); Singh et al. (2022); Radford et al. (2021) that consider dual levels.

Geometric Multimodal Contrastive Poklukar et al. (2022) processes modality data into an intermediate representation, and maps it into a latent representation space wherein the contrastive learning objective can be applied. Duan et al. Duan et al. (2022) propose a codebook-based approach to bridge between the image and its textual features, and introduce a distillation algorithm that helps unimodal and cross-modal contrastive optimization. Unlike these models, Mul2vec focuses on the global information from the underlying relationship between an entity and its modality-independent feature, its text, rather than from the text-feature pair on each entity. These representations are obtained from different inputs while sharing this space, thus ensuring that the dual-level representations are complementary.

# 3 MUL2VEC

## 3.1 OUR IDEA AND METHODOLOGY

Our intuition is that modality-specific features and corresponding text are different views of the same entity, and have common information, the global information, behind them. This information forms a modality-free space, the semantic space, so that we can directly measure their semantic similarity through their proximity, as shown in Figure 1(left). This idea ensures that similar entities have similar representations even on different levels, and allows us to explore the complementary benefits of these features. Our approach uses modality-independent data tuples (entity ID-text-words) rather than the modality-specific data pairs (text-feature) used in conventional methods.

To discover the global information, and learn semantic representations from this information, we propose a learning framework, Mul2vec, and its learning tasks. Mul2vec places semantically similar entities close together in the semantic space using their representations, while others learn different views (e.g., text and image representations) of the same entity closer together in the contextual space. Although the contextual representation is gained from other methods Singh et al. (2022); Hu & Singh (2021); Radford et al. (2021); Poklukar et al. (2022); Duan et al. (2022); Radford et al. (2021); Jia et al. (2021); Lu et al. (2022); Ma et al. (2022), the semantic representation can be obtained through combining these methods with Mul2vec, and complements the contextual representation.

**Why propose NTF to obtain semantic representations?** Variational AutoEncoder (VAE) Kingma & Welling (2014) is framework for learning multi-modal representations via latent variables. As multi-modal VAEs focus on only a subset of the modalities, Javaloy et al. Javaloy et al. (2022) refer to this limitation as modality collapse, and propose a general pipeline to enforce impartial optimization across modalities. As latent factor models describe the statistical relationships between word occurrences, combining global with local information enables us to represent words in one semantic space Shi et al. (2017). Because Mul2vec aims to learn not only target entities but also entity related elements (i.e., entity and token related tokens such as words and text), and capture their relationships simultaneously, and does not impose the Gaussian distribution constraint on the latent space as VAEs do, it employs latent factor models rather than VAE. These models enables the same entity to take on similar expressions at different levels.

As the computational cost of Non-negative Tensor Factorization (NTF) is high Shashua & Hazan (2005), Mul2vec shares a similar approach with the well-known neural technique Wu et al. (2019); He et al. (2017); Liu et al. (2019) and extends NTF to learn semantic representations under different modality-specific features.

**What are complementary relationships?** Unlike using codewords Duan et al. (2022) to quantize the joint output space, the semantic space provides a means for contrastive reasoning against other entities rather than different representations (e.g., text and features) of the same entity. By having multiple information flows in each entity, shown in Figure 1(center), the other information can complement the insufficient information and thus obtain a robust representation of the entity. These flows allow us to bridge the semantic gaps Liu et al. (2019); Kawamae (2018), bring the dual representations of the same entity closer together, and explain the relationships between entities over modalities. This is the reason why we refer to the global information as topic.

**How to design Mul2vec for dual-level alignment?** While NTF captures the high-order interactions between observations and the dependencies over sentences as latent factors, the text encoder learns dependency among the input elements as local information such as context. As they have different strengths and perspectives, combining them helps to compensate feature scarcity while learning the same data independently as achieved by the others, aligning them at the dual-level, and sharing and learning the resulting contextual and semantic representations.

As the semantic representations complement the contextual representations, and are universal over modalities, we can assign the semantic representation to an unseen entity via a topic from its text. This topic assignment means that NTF provides Mul2vec with the inductive biases. The architecture of Mul2vec ensures that each text has both the contextual representation and the semantic representation, and similar texts have high similarity in both levels, and learn them in a coordinated way using DLM, as shown in Figure 1(center).

## 3.2 PROBLEM FORMULATION AND DEFINITIONS

Given entities with their features and texts, Mul2vec learns the contextual/semantic representations of the $l$-th entity, $\mathbf{e}_l/\mathbf{g}_l$. As shown in Figure 1(center), Document ID (DID) is the identification of each entity (text or document) and is automatically assigned a random number if missing. where each document is a set/sequence of words. Like the entity, the $n$-th document and the $v$-th word have their own contextual/semantic representations, $\mathbf{e}_n/\theta_n$, and, $\mathbf{e}_v/\mathbf{t}_v$, respectively. Mul2vec initializes the contextual representation of the $l$-th entity, $\mathbf{e}_l$, and replaces it with the value that the text encoder learned from its modality-specific input using its input encoders. While Mul2vec requires a text for each entity, it accepts each entity even if a modality-specific feature is missing or partially missing texts, and outputs its final representations.

## 3.3 NEURAL TENSOR FACTORIZATION (NTF)

As we assume that there is not only a global structure of latent factors, but also a local structure of interaction in each entity, Mul2vec adopts the neural treatment of tensor factorization Wu et al. (2019); He et al. (2017); Liu et al. (2019); two pathways are used to explicitly model the interactions between documents, words, and IDs via global representations. Given the document-word-entity's ID tensor $\mathcal{D} \in \mathbb{R}^{\mathbf{N} \times V \times \mathbf{L}}$ for $N$ documents, $V$ words, and $L$ IDs, NTF decomposes this 3rd order tensor into three latent factor matrices $\Theta = \{\theta_{n,k}\} \in \mathbb{R}^{N \times K}$, $\mathbf{T} = \{\mathbf{t}_{\mathbf{v},\mathbf{k}}\} \in \mathbb{R}^{\mathbf{V} \times \mathbf{K}}$ and $\mathbf{G} = \{\mathbf{g}_{\mathbf{l},\mathbf{k}}\} \in \mathbb{R}^{\mathbf{L} \times \mathbf{K}}$, where the $k$-th row vector of each matrix corresponds to the $k$-th factor of the tensor. Following canonical polyadic decomposition Carroll & Chang (1970), $\mathcal{D}$ is factorized as:

$$\mathcal{D} \approx \sum_{\mathbf{k=1}}^{\mathbf{K}} \theta_{\mathbf{n,k}} \mathbf{t}_{\mathbf{v,k}} \mathbf{g}_{\mathbf{l,k}}, \tag{1}$$

where $\theta_{n,k}, t_{v,k}, g_{l,k}$ denotes the $k$-th value of the $n$-th document, $\theta_n$, $v$-th word, $t_v$, and $l$-th id, $g_l$. Mul2vec extends Eq (1) to the Neural Tensor Factorization (NTF) to learn semantic representations according to given contextual representations. Mul2vec incorporates Multi-Layer Perceptron (MLP) to learn their interactions, and ensemble them following the neural approach. Following the model architecture shown in Figure 1, $d_{n,v,l}$ and $\hat{d}_{n,v,l}$ denote the $n, v, l$-th observed entry in the tensor $\mathcal{D}$ and the predicted value of $d_{n,v,l}$, $d_{n \cdot v, l}$, respectively. Using Eq (1), we construct $\mathcal{D} = \{\mathbf{d}_{\mathbf{n,v,l}}\} \in \mathbb{R}^{\mathbf{N} \times \mathbf{V} \times \mathbf{L}}$. After performing weighting methods as pre-experiments and comparing their results, where we weight each entry with $\mathbf{tf} - \mathbf{idf}$ for the $v$-th word in the $n$-th document with the $l$-th ID and set it to $d_{n,v,l}$. This weighting measures $\mathbf{tf}$ and $\mathbf{idf}$ in each document, and inverse document frequency with $l$; we try some weighting methods as preliminary experiment, compare their results, and confirm that this weighting offers greater effectiveness than using only frequency and others. Note that we normalize $d_{n,v,l}$ into [0,1] to avoid gaps between texts.

According to the Neural Matrix Factorization, we define the mapping function, $\phi_{NTF}$, of NTF as:

$$\phi_{NTF}(\theta_n, t_v, g_l) = \theta_n \odot t_v \odot g_l, \tag{2}$$

where $\odot$ denotes the element-wise product of vectors.

With the increasing use of deep neural networks to capture the higher-order and non-linear interaction between features in data, complex interactions are being explored by stacking multilayer full connection layers Wu et al. (2019) rather than simple vector concatenation. Mul2vec concatenates embedding vectors of $t_v, \theta_n, g_l$ and feeds them to MLP. We can express the mapping function, $\phi_{MLP,h}$, of the $h$-th layer in MLP as follows:

$$\phi_{MLP,1}(e_n, e_v, e_l) = a_1(\mathbf{W}_1[e_n; e_v; e_l] + b_1), \phi_{MLP,2}(Z_1) = a_2(\mathbf{W}_2 Z_1 + b_2), \cdots,$$
$$\phi_{MLP,H}(Z_{H-1}) = a_H(\mathbf{W}_H Z_{H-1} + b_H), \tag{3}$$

where ; represents the concatenation operation, $\mathbf{W_h}$ and $b_h$ are the learned projection matrix and bias of the $h$-th neural layer, respectively, $a_h$ is the activation function of the perceptron of the $h$-th layer, and $H$ is the number of hidden layers indexed by $h$. As with the activation function, Mul2vec uses Rectifier (ReLU) to output quantitative values that provide multi-dimensional interactions.

By combining Eq (2) and (3)) on the last hidden layer, we can formulate the predictive function of NTF in Mul2vec as:

$$\hat{d}_{n,v,l} = \sigma(h^T \begin{bmatrix} \phi_{NTF}(\theta_n, t_v, g_l) \\ \phi_{MLP,H}(Z_{H-1}) \end{bmatrix}), \tag{4}$$

where $h^T$ denotes the edge weights of the last hidden layer; $\sigma$ is the sigmoid function. Finally, we define our objective function in the factorization procedure as:

$$L_{\mathcal{D}} = \sum_{d_{n,v,l} \in D \cup D_- = \mathcal{D}} (d_{n,v,l} - \hat{d}_{n,v,l})^2 + \lambda_\omega ||\omega||^2, \tag{5}$$

where $D$, and $D_-$ denote the set of observed entries in $\mathcal{D}$, and the set of negative entries, respectively, which can be all (or sampled) unobserved interactions, respectively; $\omega$ denotes all parameters, and $\lambda_\omega$ is used as a regularization to avoid model overfitting. As shown in Figure 1(right), NTF can be trained by minimizing this loss function between the observed interaction data, $d_{n,v,l}$, and the factorization representation, $\hat{d}_{n,v,l}$. As Adaptive Moment Estimation (Adam) Kingma & Ba (2015) can automatically tune the learning rate during training, and often provide faster convergence than the stochastic gradient descent algorithm, we use Adam via mini-batches to update parameters, and adopt the dropout strategy Srivastava et al. (2014) for network optimization.

## 3.4 TRAINING OBJECTIVES

As shown in Figure 1(center, right), Mul2vec shares the contextual representations, documents (texts) and words (tokens), with the text encoder. Mul2vec learns a semantic representation of each entity as its token embedding from the initial value, without its features (e.g., the visual feature if the entity type is image). Since the text encoder and NTF share the contextual representations, $e_n$ and $e_v$, Mul2vec takes them as initial values, and learns more complex dependencies with the NTF than text encoders of pre-trained models.

Motivated by previous models Chen et al. (2020); Su et al. (2020); Li et al. (2020a), our training objectives consist of two tasks: Linguistic Matching (ILM) and Dual Level Matching (DLM).

**ID Linguistic Matching (ILM)**: Motivated by the previous models Li et al. (2020a); Chen et al. (2020), we define ILM for NTF to learn alignments between all IDs and the text at the instance-level rather than at the token/ID-level. Given an entity and its document as ID and DID, the objective of this task is to predict whether this document semantically matches the entity. As similar entities could have similar representation vectors, we employ a triplet objective-based function to evaluate their similarities in the semantic representation space. We refer to the semantic representation of the $l$-th ID, and its document as an anchor vector $\mathbf{v_a}$, and the positive vector $\mathbf{v_p}$, respectively, and call that of the other document, negative vector $\mathbf{e_n}$. Given these vectors, triplet loss tunes the model so that the distance between $\mathbf{v_a}$ and $\mathbf{v_p}$ is less than the distance between $\mathbf{v_a}$ and $\mathbf{v_n}$. The objective here is to minimize the following loss function:

$$\mathcal{L}_{ILM}(\zeta) = \max_{(\mathbf{v_a}, \mathbf{v_p}, \mathbf{v_n}) \sim \mathbf{B}} (||\mathbf{v_a} - \mathbf{v_p}|| - ||\mathbf{v_a} - \mathbf{v_n}|| + \epsilon, \mathbf{0}), \tag{6}$$

where $\mathbf{B}$ is each batch, and $\mathbf{v_n}$ is in the same batch, $|| \bullet ||$ is a distance metric and $\epsilon$ is the margin that ensures that $\mathbf{e_p}$ is at least $\epsilon$ closer to $\mathbf{e_a}$ than $\mathbf{e_n}$.

**Dual Level Matching (DLM)**: The objective of DLM is to minimize the distance between the semantic representation and the contextual representation, as these representations are obtained from the same entity through different inputs. In other words, texts with similar content have similar representation vectors in the dual representations, and their distance is more smaller than that of other text pairs, because they have more words or tokens in common than the others. Like the ILM, we employ a triplet objective-based function to evaluate their distance in the dual-level representations, where we compare it with the contrastive function Khosla et al. (2020); Radford et al. (2021) in pre-experiment and select the former function. We refer the contextual representation of $n$-th DID, and its semantic representation, to an anchor vector $\mathbf{v_a}$, and positive vector $\mathbf{v_p}$, respectively, and call the semantic representation of the other document, negative vector $\mathbf{e_n}$. Given these vectors, triplet loss tunes the model such that the distance between $\mathbf{v_a}$ and $\mathbf{v_p}$ is smaller than the distance between $\mathbf{v_a}$ and $\mathbf{v_n}$. The objective here is to minimize the following loss function:

$$\mathcal{L}_{DLM}(\zeta) = \max_{(\mathbf{v_a}, \mathbf{v_p}, \mathbf{v_n}) \sim \mathbf{B}} (||\mathbf{v_a} - \mathbf{v_p}|| - ||\mathbf{v_a} - \mathbf{v_n}|| + \epsilon, \mathbf{0}), \tag{7}$$

where $\mathbf{B}$ is each batch, and $\mathbf{v_n}$ is in the same batch, $|| \bullet ||$ is a distance metric and $\epsilon$ is the margin that ensures that $\mathbf{e_p}$ is at least $\epsilon$ closer to $\mathbf{e_a}$ than $\mathbf{e_n}$.

Table 1: Basic statistics of the datasets used in this paper: D, V, L and I denote #documents, #vocabulary, #IDs, and #Images, respectively

| data | category | D | V | L | I |
|---|---|---|---|---|---|
| Flickr30 | file name | 31,783 | 8,511 | 31,783 | 31,783 |
| MSCOCO | file name | 581,286 | 17,931 | 116,195 | 116,195 |

## 3.5 FINE-TUNING

In Mul2vec, NTF can work with pre-trained multi-modal models with simple modifications to the properly formatted input/output, the loss functions, DLM and ILM. This framework can be fine-tuned to suit particular downstream multi-modal tasks. Overall, we have three training regimes corresponding to the ID-text inputs Our final training objective is the sum of the above losses , Eq (5), (6), (7):

$$\mathcal{L} = L_{\mathcal{D}} + \lambda_{ILM}\mathcal{L}_{ILM}(\zeta) + \lambda_{DLM}\mathcal{L}_{DLM}(\zeta), \tag{8}$$

where $\lambda_{ILM}$ and $\lambda_{DLM}$ are used as regularization to avoid model overfitting. While the objective functions depend on pre-trained models, the training objectives are to optimize Eq (8) over the training data while freezing the parameters of these models.

## 4 EXPERIMENTS

### 4.1 DATASETS, SETUP AND DESIGN

**Datasets** We conducted evaluations on Flickr30k [1], and MSCOCO [2], as both are publicly available, and are widely used in multi-modal search studies Wang et al. (2019); Li et al. (2020a) for modalities of texts and images (entity). For Flickr data, we follow the setting of Karpathy & Li (2015) and split its word tokens to gain training/validation/test datasets. Their statistics are shown in Table 1. That is, the size of ID and words in NTF matches the size of product/image ID and tokenizer used in Transformer models.

**Experimental Setup** We implemented our model using Pytorch 2.0[3] and horovod 0.21.0[4]; these codes will be released later. All models were trained on 4 V100 GPUs with 32G memory. In these experiments, we set the #layers of the Neural Tensor Factorization network to 3, and work Mul2vec with pre-trained Transformer-based multi-modal models. As a common setting for both, we set the dimension of embedding, and the maximum length of input sequence to 768, and 512, respectively, and set many parameters of Mul2vec to those of other models for fair comparison.

As with training NTF, we used both backpropagation and stochastic gradient descent (SGD) for optimization, and employed Adaptive Moment Estimation (Adam) Kingma & Ba (2015) via mini-batch with 256 size for parameter update; the dropout strategy Srivastava et al. (2014) was used with 0.2 rate to prevent overfitting.

As with training Transformer, we ran the models for 20 epochs using Adam with $\beta1 = 0.9$, $\beta2 = 0.999$ for optimization over mini-batches for parameter update and adopted the dropout strategy Srivastava et al. (2014) to optimize networks. The learning rate was 3e-5, with linear warmup over the first 100 steps and linear decay, where we set the dropout rate, the weight decay, and the batch size to 0.1, 0.01, and 256, respectively. The pre-training procedure was limited to about 1,000 steps. As Mul2vec incorporates the NTF network and adds new inputs to the original multi-modal models, we randomly initialized their parameters from a Gaussian distribution with mean of 0 and standard deviation of 0.02. As base pre-trained multi-modal models, we select UNITER, VL-BERT, VD-BERT, ViT, Oscar, CLIP, BLIP; note that their parameters were optimized over the conceptual caption dataset Sharma et al. (2018) for fair comparison. This dataset contains 3.3M image and caption pairs, and is popular for cross-modal pre-training.

---

[1]http://bryanplummer.com/Flickr30kEntities/

[2]https://cocodataset.org/#home

[3]https://pytorch.org/

[4]https://github.com/horovod/horovod

Table 2: Qualitative results of (top) T2I obtained, (bottom) algebraic operations over image and words by Mul2vec: In scenario 2 (top), the answer image is associated with query sentence in the test data.

| Query (words) | Answer image | Answer from Mul2vec |
|---|---|---|
| A black and white dog are running in a grassy garden surrounded by a white fence |  |  |
| **Query (image - word)** | | **Answer from Mul2vec** |
|  - grass | |  |
|  - dog | |  |

**Experiment design** To verify the representation learning ability of Mul2vec, which learns semantic representations without entity features.

## 4.2 MULTI-MODAL SEARCH USING PRE-TRAINED MODELS

When both words and images are embedded in the same space, we can perform algebraic operations over them as shown in Table 2, where we formed queries by combining an image and two words; the top 3 images are shown according to the cosine similarity between this query's embedding vector and the images' embedding vector. Although this dataset contains many dog images, this table shows that Mul2vec can (1) find, through them, images of dogs that are similar in color and behavior to the query by similarity by using embedded representations instead of keyword matching, without visual features, and (2) map the source data into the same semantic space without visual features. This allows us to perform algebraic operations over images and words. Since we gained similar results to the keyword matching method in this experiment setting, we could confirm that Mul2vec treated grassy, garden, and field as synonyms of grass, and ranked images by cosine similarity between the results of the operation and the images. The reason for this result is that even between data that appear to be fragmented, if the latent factors exist over these datasets, their representations can be shared across the latent space, improving the quality of their representations, embedding.

To measure how much Mul2vec can contribute to state-of-the-art pre-trained vision-language (V-L) models in multi-modal search, we compare its contribution and show the resulting text/image task quality metrics of I2T and T2I in Table 3. We use UNITER Chen et al. (2020),VL-BERT Su et al. (2020), VD-BERT Wang et al. (2020), ViT Dosovitskiy et al. (2021), Oscar Li et al. (2020b), CLIP Radford et al. (2021), and BLIP Li et al. (2022) as pre-trained models, where we reuse a prior implementation[5]; we follow their work and find better hyper-parameters in fine-tuning over the Flickr30 and MSCOCO datasets. We plug Mul2vec into these models, and train only Mul2vec over these datasets, where the V-L models are frozen. To evaluate the effect of semantic representations, we also prepared the original data and the data with 20% of each of these images and texts randomly masked, **incomplete data**, and compared the improvement obatained over the models without Mul2vec. That is, we compare the effects of using the semantic representation of $n$-th text, $\theta_n$, instead of the corresponding contextual representation, $e_n$, the output of each V-L model.

---

[5]https://github.com/ChenRocks/UNITER, https://github.com/jackroos/VL-BERT, https://github.com/Wangt-CN/MTFN-RR-PyTorch-Code, https://github.com/salesforce/VD-BERT, https://github.com/jeonsworld/ViT-pytorch, https://github.com/microsoft/Oscar, https://github.com/openai/CLIP, https://github.com/salesforce/BLIP

Table 3: Contributions of Mul2vec over (upper)Flickr30, and (lower) MSCOCO: In these experiments, we apply the semantic representation of text (document), $\theta_n$, instead of the contextual representation to these tasks, while we use the contextual representation of each image that has been gained from each vision-language model.

| models | I2T(%) | | | T2I(%) | | |
|---|---|---|---|---|---|---|
| | R@1 | R@5 | R@10 | R@1 | R@5 | R@10 |
| UNITERChen et al. (2020) | +1.2 | +3.3 | +6.1 | +1.7 | +3.5 | +6.8 |
| VL-BERT Su et al. (2020) | +1.4 | +3.5 | +6.4 | +1.7 | +3.6 | +6.8 |
| VD-BERT Wang et al. (2020) | +1.1 | +3.1 | +6.2 | +1.2 | +3.1 | +6.4 |
| ViT Dosovitskiy et al. (2021) | +1.1 | +3.2 | +6.4 | +1.3 | +3.3 | +6.2 |
| Oscar Li et al. (2020b) | +1.0 | +3.0 | +6.5 | +1.2 | +3.2 | +6.5 |
| CLIP Radford et al. (2021) | +1.0 | +3.0 | +6.6 | +1.1 | +3.4 | +6.4 |
| BLIP Li et al. (2022) | +1.2 | +3.1 | +6.3 | +1.1 | +3.0 | +6.6 |
| UNITERChen et al. (2020) | +5.3 | +9.8 | +15.7 | +5.7 | +10.2 | +17.4 |
| VL-BERT Su et al. (2020) | +5.8 | +10.4 | +16.2 | +6.2 | +11.5 | +17.8 |
| VD-BERT Wang et al. (2020) | +5.1 | +9.4 | +15.5 | +5.5 | +10.4 | +17.0 |
| ViT Dosovitskiy et al. (2021) | +5.2 | +9.2 | +14.8 | +5.3 | +10.1 | +17.2 |
| Oscar Li et al. (2020b) | +5.1 | +9.1 | +14.6 | +5.2 | +10.2 | +17.5 |
| CLIP Radford et al. (2021) | +5.1 | +9.1 | +14.3 | +5.1 | +10.3 | +17.4 |
| BLIP Li et al. (2022) | +5.1 | +9.3 | +14.4 | +5.1 | +10.2 | +17.3 |

Table 4: Ablation analysis over Flickr30: The bold value denotes the statistical significance for $p < 0.01$ with the student t-test, compared to the best baseline. $K$ is the number of topics. In ILM and DLM, the value denotes $\lambda_{ILM}$ and $\lambda_{DLM}$. In prediction, we use Mean Absolute Error (MAE) to evaluate the difference between $\hat{d}_{n,v,l}$ (Eq (4)) and actual value, $d_{n,v,l}$.

| NTF | | | Tasks | | I2T(R@N) | T2I(R@N) | prediction |
|---|---|---|---|---|---|---|---|
| TF | MLP | K | ILM | DLM | @1, @5,@10 | @1,@5, R@10 | MAE |
| w/ | w/ | 50 | 0.1 | 0.1 | **69.1, 93.1, 98.3** | **60.7, 79.8, 91.6** | 0.10 |
| w/ | w/ | 40 | 0.1 | 0.1 | 67.9, 91.3, 97.2 | 58.3, 78.4, 89.9 | 0.10 |
| w/ | w/ | 30 | 0.1 | 0.1 | 67.8, 91.2, 97.1 | 58.1, 78.2, 89.9 | 0.10 |
| w/ | w/ | 20 | 0.1 | 0.1 | 67.6, 88.4, 92.3 | 56.2, 74.7, 86.5 | 0.10 |
| w/ | w/ | 10 | 0.1 | 0.1 | 67.2, 88.1, 90.4 | 55.5, 74.3, 83.3 | 0.10 |
| w/ | w/ | 20 | 0.0 | 0.1 | 67.1, 86.3, 88.7 | 54.6, 73.8, 82.7 | 0.10 |
| w/ | w/ | 20 | 0.1 | 0.0 | 65.8, 85.9, 84.4 | 52.5, 70.2, 80.3 | 0.11 |
| w/ | w/ | 20 | 0.2 | 0.1 | 67.8, 89.9, 96.8 | 56.6, 77.2, 89.8 | 0.10 |
| w/ | w/ | 20 | 0.1 | 0.2 | 68.2, 91.9, 97.3 | 59.8, 78.5, 90.8 | 0.10 |
| w/ | w/o | 20 | 0.1 | 0.1 | 66.1, 86.3, 85.1 | 53.2, 70.3, 80.6 | 0.13 |

Although the task is limited, Table 3 shows that the contribution of the semantic representation to pre-trained V-L, multi-modal, models is higher than the contextual representation for these tasks. This result implies that NTF guesses the missing values by topics, and helps Mul2vec to complement pre-trained models with semantic representations comparable to their features and learn better representations than was possible by using Transformer alone; Mul2vec helps to offsetting the missing modalities Chen & Zhang (2020); Ma et al. (2021). A manual error analysis shows that some cases marked as errors were in fact correctly judged if we allow partial matching of words in a document or objects in an image. In practice, these judgments were often incorrect if the sentence was long or when there were many objects in the image.

## 4.3 ABLATION ANALYSIS

To verify the effectiveness of the proposed components and tasks of Mul2vec, we conduct an ablation analysis, in which Mul2vec works with BLIP. As Mul2vec needs TF to learn the semantic representation, TF cannot be excluded from this analysis. We set up experiments to explore the possible combinations, use the contextual representation of image and the semantic representation of the text, $\theta_n$, and show their performance on the Flickr30, and MSCOCO in Table 4, and Table 5, respectively. As in the previous experiments, we conduct this analysis under the incomplete data. Table 4 and Table 5 show that 1) Mul2vec requires both TF and MLP to learn the semantic represen-

Table 5: Ablation analysis over MSCOCO: The bold value, $K$, $\lambda_{ILM}$, $\lambda_{DLM}$, and MAE are the same as the definition in Table 4

| NTF | | | Tasks | | I2T(R@N) | T2I(R@N) | prediction |
| TF | MLP | K | ILM | DLM | @1, @5,@10 | @1,@5, R@10 | MAE |
|---|---|---|---|---|---|---|---|
| w/ | w/ | 50 | 0.1 | 0.1 | **67.9, 90.2, 95.5** | **57.8, 77.4, 89.3** | 0.10 |
| w/ | w/ | 40 | 0.1 | 0.1 | 65.6, 89.4, 94.2 | 56.3, 76.1, 87.5 | 0.10 |
| w/ | w/ | 30 | 0.1 | 0.1 | 65.3, 86.1, 90.5 | 56.1, 74.7, 86.5 | 0.11 |
| w/ | w/ | 20 | 0.1 | 0.1 | 65.2, 85.2, 90.2 | 55.2, 74.2, 86.3 | 0.11 |
| w/ | w/ | 10 | 0.1 | 0.1 | 64.8, 84.8, 89.8 | 54.6, 73.8, 85.5 | 0.11 |
| w/ | w/ | 20 | 0.0 | 0.1 | 64.2, 84.4, 86.3 | 52.3, 72.5, 80.6 | 0.11 |
| w/ | w/ | 20 | 0.1 | 0.0 | 63.5, 83.2, 82.2 | 50.3, 67.8, 78.4 | 0.12 |
| w/ | w/ | 20 | 0.2 | 0.1 | 65.5, 87.4, 93.5 | 54.2, 75.6, 87.1 | 0.11 |
| w/ | w/ | 20 | 0.1 | 0.2 | 67.1, 89.1, 94.8 | 56.9, 76.8, 88.7 | 0.10 |
| w/ | w/o | 20 | 0.1 | 0.1 | 64.2, 83.9, 84.6 | 52.3, 69.7, 79.1 | 0.12 |

Table 6: Runtime comparison in fine-tuning over (upper)Flickr30, and (lower) MSCOCO

| UNITER | VL-BERT | VD-BERT | ViI | Oscar | BLIP |
|---|---|---|---|---|---|
| +1.1 | +1.2 | +1.2 | +1.1 | +1.2 | +2.1 |
| +1.4 | +1.5 | +1.8 | +1.6 | +1.6 | +2.8 |

tation, 2) the quality of the semantic representation improves in proportion to the number of topics, and 3) both ILM and DLM are essential training tasks for our goal, while DLM has a greater impact on learning the semantic representation than ILM.

### 4.4 RUNTIME ANALYSIS

The rate of increase in computational cost incurred when Mul2vec cooperates with the models shown in Table 3, is given in Table 6. NTF shares the representations between MLP and Transformer, and thus requires fewer parameters in total than the one block of Transformer encoder, where their learning also uses different paths (MLP/Transformer and TF/Transformer). Although the computational cost of NTF appears to be high, we can compute it at a similar cost to other latent variable models, VAEs, through batching and parallelization. Table 6 confirms that Mul2vec is effective in accelerating their convergence, and thus reducing computational complexity.

## 5 DISCUSSION AND LIMITATIONS

While Mul2vec does not aim to discover topics, directly, they affect the quality of semantic representations. We compare Mul2vec with NTF, NMF and related models Xun et al. (2017) over the Amazon dataset using the topic coherence measure Mimno et al. (2010) which compares topic models based on their human-interpretability; a higher topic coherence score indicates higher topic quality. The comparison using the paired $t$-test shows that Mul2vec achieves higher scores, suggesting that it offers superior interpretability compared to the other models. The reason why topics are resistant to missing values may be because they can be shared across entities. Limitations of Mul2vec are not effective when there are many missing data.

## 6 CONCLUSION

Based on the idea that modality-specific features and the corresponding text are different views of the same entity, we propose a framework, Mul2vec, for learning semantic representations as modality-free representations that can align various modal-type entities. For Mul2vec, we extend the tensor factorization to NTF, which can cooperate with pre-trained multi-modal models or text encoders, and learn the semantic representation from a given data and the contextual representation, and propose objectives that learn modality-free representations via ILM and DLM in the multi-task learning process. Unlike other models, Experiments show that Mul2vec aligns images and text on the dual-level representations and can contribute to pre-trained multi-modal models that suit downstream tasks under incomplete data.

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
