# Supplementary material: Learning Multi-modal Representations Under Incomplete Data Via Dual Level Alignments

## 1 EXPERIMENTS

### 1.1 DATASETS, SETUP AND DESIGN

**Datasets** We conducted evaluations on the Amazon review data sets (Amazon) [1], Flickr30k [2]. We use Amazon for evaluating different modalities of texts and products (entity). For Amazon, we selected two interpretable categories, counted the number of reviews per product, and used the reviews of the top 103 and 229 products (items) in Movie and CD categories, respectively, where each product had at least 100 reviews. This data size allows the baselines Yang et al. (2016); Guoyin Wang & etc (2018) to run on non-GPU machines without substantial modification for reproduction; #IDs is the upper number permitting subjective evaluation. For non-Transformer based models, we used NLTK [3] using ¡UNK¿ to replace the stop words and rare words (frequency lower than 5 in each collection); all words were converted to lowercase. Finally, we used asin product-ID (used in the data set) as IDs for the Amazon data, where each ID can be converted into a product name by using a product name look-up table; the name is used only for interpretation of the results.

### 1.2 ALGEBRAIC OPERATIONS OVER IDS

Mul2vec(-) learns common representations that enable users to perform semantic algebraic operations on different types such as IDs and words, and we show examples of the algebraic operations possible in Table 2. For the popular items in each dataset, we asked experienced raters to select operations that could be readily interpreted by people unfamiliar with movies as examples. Since they involve famous products, we can interpret the semantic proximity of these operations' results. The first row of this table shows that subtracting "History" (word) from "Gladiator" (item) implies that the closest semantic film is "The Dark Knight", and the word "swordsman". In fact, "Gladiator" is the story of a "swordsman" and its time setting is different from "The Dark Knight". More reviews are needed to get better clarity and interpretability.

Each ID identifies, depending on the entity type, a unique entity, as shown in Figure **??**. In fact, less than half of the entries in this data, explicitly mention the product name and category. As each type has intrinsically different scale and feature representations (e.g., images vs. movies), pre-processing requires type-specific normalization in order to prevent learning bias. While entity-specific approaches Chalkidis et al. (2019); Ren et al. (2017); Zhou et al. (2019) treat only those entities having a string ID, our utilization of IDs allows Mul2vec to learn multi-modal embeddings over a greater variety of non-verbal data types.

---

[1] http://jmcauley.ucsd.edu/data/amazon/qa/
[2] http://bryanplummer.com/Flickr30kEntities/
[3] NLTK 3.2.4: http://www.nltk.org

Table 1: Basic statistics of the datasets used in this paper: D, V, L and I denote #documents, #vocabulary, #IDs, and #Images, respectively

| data | category | D | V | L | I |
|------|----------|-----|-----|-----|-----|
| Amazon | Movies and TV | 9,986 | 45,155 | 103 | N/A |
| | CDs and Vinyl | 22,226 | 56,460 | 229 | N/A |
| Flickr30 | file name | 31,783 | 8,511 | 31,783 | 31,783 |
| NSCOCO | file name | 581,286 | 17,931 | 116,195 | 116,195 |

Table 2: Most similar entities detected by Mul2vec(-) from Amazon Movie. In this operation, cosine similarity was used to select the 4 items or words closest to the corresponding operation. The string highlighted in blue is the title of the movie that corresponds to the movie ID.

| Operation Type | example of operation | similar items | similar words |
|------|------|------|------|
| item - word | Gladiator - history | The Dark Knight, Batman Begins, Spider Man 2, The Incredibles | swordsman, roma, battle, empire |
| item + word | Gladiator + future | Independence Day, The Matrix, Star Trek The Original Series, Wall-E | technology, machine, metal, technically |
| item - item | Gladiator - Mary Poppins | Terminator 2, Memento, Shrek, Halloween | idiotic, destroying, trouble, tranquillity |
| item + item | Gladiator + Mary Poppins | The Lord of the Rings, Jurassic Park, Superman Returns, The Sound of Music | effect, popular, extraordinary, family |

Table 3: Performance of image-text matching tasks over Flickr30: The best results are marked in bold, where the bold value denotes the statistical significance for $p < 0.01$, compared to the best baseline.

| Method | I2T | | | T2I | | |
|------|------|------|------|------|------|------|
| | R@1 | R@5 | R@10 | R@1 | R@5 | R@10 |
| Doc2Vec Le & Mikolov (2014) | 12.5 | 21.7 | 34.3 | 6.2 | 8.4 | 15.5 |
| HAN Yang et al. (2016) | 14.4 | 24.8 | 38.7 | 8.7 | 11.5 | 19.6 |
| LEAM Guoyin Wang & etc (2018) | 14.6 | 23.5 | 36.3 | 8.9 | 11.8 | 20.1 |
| BERT Devlin et al. (2019) | 22.9 | 32.4 | 47.7 | 18.5 | 22.8 | 34.1 |
| Mul2vec(+BERT) | **66.8** | **86.5** | **94.4** | **52.5** | **76.2** | **87.3** |