# OpenReview forum: "Learning Multi-modal Representations Under Incomplete Data Via Dual Level Alignments"
_ICLR.cc/2024/Conference — Submitted to ICLR 2024_

### Official Review · Reviewer_ouCz · 2023-10-23

**Soundness:** 2 fair
**Presentation:** 1 poor
**Contribution:** 2 fair
**Rating:** 3
**Confidence:** 3

**Summary:**

Their goal is to learn modality-free representations of a wide variety of entity types, that can be applied to multi-modal tasks under incomplete data. They propose Mul2vec framework which consists of NTF (Neural Tensor Factorization), and training objectives, DLM ( Dual Level Matching ) and ILM (Linguistic Matching). Mul2vec aligns contextual and semantic representations and optimizes them. Experiments show that Mul2vec outperforms baselines in downstream tasks.

**Strengths:**

Experiments show that Mul2vec outperforms baselines in downstream tasks.

**Weaknesses:**

The writing makes it very difficult for readers to understand the paper.
1. The entities, IDs, DIDs are not clearly defined.
2. It seems the method section (3.3, 3.4) mostly focus on the text modality. However, how it works with multiple modalities and how it works with pre-trained multi-modal models are not clear.  The paper only mentions "with simple modifications to the properly formatted input/output, the loss functions, DLM and ILM". More details are needed.

**Questions:**

1. It seems the model learns 2 sets of embeddings, e (3.4) and g (3.3). Are there any relationships between the the two sets of embeddings or they are independent? Which set is used for evaluation in Table 2? Why image uses e and text uses g during retrieval?
2. The supplement references a figure, but there is no figure in the document.

---

> ### Author Response · Authors · 2023-11-12
> **Rebuttal for Reviewer ouCz**
>
> Thank you for reviewing our paper.
> We'd like to respond to the comments here.
>
> ## Weaknesses:
> **1) The entities, IDs, DIDs are not clearly defined.**\
> We have defined these many times in the text as follows, but do you need more specific examples?
> \
> **Entities:** *``entity types (e.g., text, image, object)* in abstract, because this is public defintion (e.g., each image), so we are not defining it here''*.
> \
> **IDs:** *``DID, and ID are the identifier of document, and entity, respectively''* in Figure 1 caption.
> \
> **DIDs:** *Document ID (DID) is the identification of each entity (text or document) and is automatically assigned a random number if missing *in 3.2
> \
> Specific examples of each are also presented in 4.1 and Supplementary Material.
> \
> *``We conducted evaluations on Flickr30k 1, and MSCOCO 2, as both are publicly available, and are widely used in multi-modal search studies Wang et al. (2019); Li et al. (2020a) for modalities of texts and images (entity).''* in 4.1
> \
> *``each ID can be converted into a product name by using a product name look-up table; the name is used only for interpretation of the results.''* in Supplementary Material
> \
> \
> **2) It seems the method section (3.3, 3.4) mostly focus on the text modality. However, how it works with multiple modalities and how it works with pre-trained multi-modal models are not clear. The paper only mentions "with simple modifications to the properly formatted input/output, the loss functions, DLM and ILM". More details are needed.**
> \
> As explained in Figure 1 and 3.1, *our intuition is that modality-specific features and corresponding text are different views of the same entity, and have common information, the global information, behind them*, and focuses on the text modality,
> where this  intuition *ensures that similar entities have similar representations even on different levels,
> and allows us to explore the complementary benefits of these features*.
> To discover the global information, and learn semantic representations from this information,
> Mul2vec accepts *``modality-independent data tuples (entity ID-text-words) rather than the modality-specific data pairs (text-feature) used in conventional methods''* as its input, and aligns them and optimizes the semantic representations with the corresponding contextual representations using ILM and DLM.
>
> ## Questions:
> **1) It seems the model learns 2 sets of embeddings, e (3.4) and g (3.3). Are there any relationships between the the two sets of embeddings or they are independent? Which set is used for evaluation in Table 2? Why image uses e and text uses g during retrieval?**
> \
> As explained in 3.5 and equation (8), these sets are learning to complement each other because of they are different representations of the same entity, document, and word, at the dual (contextual and semantic) levels.
> Thus, if we get both, we use them combined; otherwise, we use only the expressions we get.
> Like transformer based V-L models tell us $e_{\*}$, NTF informs us $g_{\*}$ to a given ID (entity), DID (document, text), and word.
> \
> \
> **2) The supplement references a figure, but there is no figure in the document.**
> \
> We put the figures in the supplement references because of it's consistency, priority and space in the paper.

---

> ### Comment · Reviewer_ouCz · 2023-11-23
>
> Thanks for the reply. After carefully reading your replies and other reviewers' comments, I find that multiple reviewers have raised the concern that the writing is confusing. There will be a lot of revisions needed to make the paper clear in terms (e.g. entities, IDs, documents, texts) and easy to follow. Thus, I keep my evaluation ratings.

---

> ### Author Response · Authors · 2023-11-23
> **Reply to Reviewer ouCz**
>
> Thanks for the response.
>
> Reviews would be based more on the paper, its response and **own knowledge** than other reviews.

---

### Official Review · Reviewer_beEN · 2023-10-30

**Soundness:** 1 poor
**Presentation:** 1 poor
**Contribution:** 1 poor
**Rating:** 3
**Confidence:** 2

**Summary:**

This paper seems to be proposing a method to learn a “modality-free” multi-modality representations.

**Strengths:**

This paper seems to be proposing a method to learn a “modality-free” multi-modality representations.

**Weaknesses:**

Since the beginning of this paper, its presentation confuses me a lot. For example, “Our approach uses modality-independent data tuples (entity ID-text-words) rather than the modality-specific data pairs (text-feature) used in conventional methods.” After  reading the full paper, I still don’t understand why the entity ID can be used to calculate the representations since IDs can be completely random. And I also don’t understand why “text” and “words” are different modalities and what are their different roles in this tuple.

Therefore, I completely couldn’t get the two “newly” proposed losses. Maybe other reviewers can shed some light on these confusions of mine.

**Questions:**

Since the beginning of this paper, its presentation confuses me a lot. For example, “Our approach uses modality-independent data tuples (entity ID-text-words) rather than the modality-specific data pairs (text-feature) used in conventional methods.” After  reading the full paper, I still don’t understand why the entity ID can be used to calculate the representations since IDs can be completely random. And I also don’t understand why “text” and “words” are different modalities and what are their different roles in this tuple.

Therefore, I completely couldn’t get the two “newly” proposed losses. Maybe other reviewers can shed some light on these confusions of mine.

---

> ### Author Response · Authors · 2023-11-12
> **Rebuttal for Reviewer beEN**
>
> Thank you for reviewing our paper. We'd like to respond to the comments here.
>
> # Weaknesses and Questions:
> \
> **Since the beginning of this paper, its presentation confuses me a lot. For example, “Our approach uses modality-independent data tuples (entity ID-text-words) rather than the modality-specific data pairs (text-feature) used in conventional methods.” After reading the full paper, I still don’t understand why the entity ID can be used to calculate the representations since IDs can be completely random.**
> \
> Since ID is a type of token, it gains its representation in the same way when the initial values of other type tokens (tokenized text) are updated by learning Transfomer based models.
> \
> \
> **And I also don’t understand why “text” and “words” are different modalities and what are their different roles in this tuple.**
> \
> Because text is a sequence of words, and words also occur in different sequences, their roles and the representations obtained in learning are different.
> For example. the representation of a text does not necessarily match the sum of the representations of the words it contains.

---

### Official Review · Reviewer_goSd · 2023-10-31

**Soundness:** 2 fair
**Presentation:** 3 good
**Contribution:** 2 fair
**Rating:** 3
**Confidence:** 5

**Summary:**

The paper's aim is to develop modality-independent representations of a wide range of entity types (e.g., text, image, object) that can be applied to multi-modal tasks with incomplete data (e.g., noisy data or missing modality information).
Traditional methods train models on modality-specific features (e.g., image features via visual encoding) and decode them into contextual representations of specific modalities (e.g., images and text).
Mul2vec leverages the concept that text and its associated features are two different perspectives of the same entity. It does not require modality-specific features and instead learns semantic representations.
The idea is that similar entities should have similar representations on both a contextual and semantic level, and Mul2vec aligns them and optimizes the semantic representations with the corresponding contextual representations.
Experiments have demonstrated that Mul2vec is successful in acquiring semantic representations and can be utilized to enhance pre-trained models for downstream tasks with incomplete information.

**Strengths:**

The paper proposed to tackle an important multimodal learning topic which is dealing with noisy/missing data while training.
The method demonstrate performance gain on existing SOTA VL methods and visualize the learned representation can perform subtraction in semantic across modalities.

**Weaknesses:**

(1) The writing can be improved. A lot of terminology is left undefined when presented
The NTF was mentioned multiple times but never explained since it can refer to Non-negative Tensor Factorization and Neural Tensor Factorization. Also, the I2T and T2I weren't explained. I suggest it be text-to-image retrieval.
(2) The difference between contextual information and semantic information is not clear.
Why is the proposed model learning semantics? Does semantic refer to global information from the TF-IDF document information?
From my understanding of Figure 1, semantics was learned by TF-IDF document-level information, which is a form of text-only information.
(3) The main component of Neural Tensor Factorization was proposed in previous work [Wu et al. (2017)].
Also, the training objective of ILM and DLM was from previous works, which weakens the contribution of this work.
(4) Although it claims to deal with missing modalities, it wasn't further explored by additional experiments (showing the effect of dealing with different levels of data missing, and the model is robust to it.)
(5) In the experiment, as in tables 3 and 6, only the difference was presented in the table, which makes it hard for the reader to have a sense of whether the performance is good or bad.

**Questions:**

Please answer the question regarding semantic learning in the weakness
Also, it is hard to understand the contribution of this paper. It will be better to state it clearly.

---

> ### Author Response · Authors · 2023-11-12
> **Rebuttal for Reviewer goSd**
>
> Thank you for reviewing our paper. We'd like to respond to the comments here.
>
> ## Weaknesses:
> **1) The writing can be improved. A lot of terminology is left undefined when presented The NTF was mentioned multiple times but never explained since it can refer to Non-negative Tensor Factorization and Neural Tensor Factorization. Also, the I2T and T2I weren't explained. I suggest it be text-to-image retrieval.**
> \
> We define NTF in *3.3 NEURAL TENSOR FACTORIZATION (NTF)*.
> I2T and T2I are as you indicated, and we use the, without any other explanation learned because these terms are common in V-L tasks and multi-modal domain.
> \
> \
> **(2) The difference between contextual information and semantic information is not clear. Why is the proposed model learning semantics? Does semantic refer to global information from the TF-IDF document information? From my understanding of Figure 1, semantics was learned by TF-IDF document-level information, which is a form of text-only information.**
> \
> The contextual representation is obtained by the existing model from both text and features,
> while the semantic representation is obtained from only text for the first time by Mul2vec.
> As shown in Figure 1 and stated in 3, Mul2vec learns semantics from the global information via NTF, like latent factor models (e.g., VAEs and Tensor Factorization) describe the statistical relationships between word occurrences, combining global with local information enables us to represent words in one semantic space
> As stated in 3.1,
> *``we try some weighting methods as preliminary experiment, compare their results, and confirm that this weighting (TF-IDF) offers greater effectiveness than using only frequency and others.''*
> TF-IDF is form of text-only information as you point out, and we think it is an excellent initial value for learning semantics from the global infromation as used in other models.
> \
> \
> **(3) The main component of Neural Tensor Factorization was proposed in previous work [Wu et al. (2017)]. Also, the training objective of ILM and DLM was from previous works, which weakens the contribution of this work.**
> \
> We have the same name as **Neural Tensor Factorization**, but it differs from  [Wu et al. (2019)] in 1) learning the dual level representations, 2) employing different layers (ie., TF and MLP) as its architecture, and 3) learning itself with Transformer based models.
> The form within objective function ILM and DLM is indeed an existing one,
> but we are not proposing these forms.
> From alternative candidates,
> we are selecting, modifying, validating and proposing them as the form that meets our objectives function.
> \
> \
> **(4) Although it claims to deal with missing modalities, it wasn't further explored by additional experiments (showing the effect of dealing with different levels of data missing, and the model is robust to it.)**
> \
> To test the robustness to missing data in the local information (e.g., text and image)
> we created missing data and perform experiments, *``To evaluate the effect of semantic representations, we also prepared the original data and the data with 20% of each of these images and texts randomly masked, incomplete data, and compared the improvement obatained over the models without Mul2vec.''*, as shown in 4.2.
> Can you tell us what other validations or missing data we should have considered?
> \
> \
> **(5) In the experiment, as in tables 3 and 6, only the difference was presented in the table, which makes it hard for the reader to have a sense of whether the performance is good or bad.**
> \
> Because we believe that relative improvement value is more reliable and easier for readers to evaluate than absolute values, we show these values.
> We fear that absolute values would be unreliable and unrepeatable because these values would change with the environment and other factors.
>
> ## Questions:
> **Please answer the question regarding semantic learning in the weakness Also, it is hard to understand the contribution of this paper. It will be better to state it clearly.**
> \
> Please let us know if there are any shortcomings or new issues in the above answers.

---

> > ### Comment · Reviewer_goSd · 2023-12-05
> >
> > I have read through the reviewers and rebuttals.
> > Besides the writing issue mentioned by other reviewers, the ability to deal with missing modalities is also questionable (also raised by reviewer ouCz).
> > It will be interesting to train with the missing actual modalities, such as video/audio/text pairs where some data doesn't have audio, video, or text.
> > An example will be training with HowTo100M, where some videos lack audio or ASR (text).
> > I'll keep my rating.

---

### Official Review · Reviewer_BQ2D · 2023-11-01

**Soundness:** 2 fair
**Presentation:** 1 poor
**Contribution:** 2 fair
**Rating:** 3
**Confidence:** 4

**Summary:**

This paper proposes a framework for learning multi-modal representations under incomplete data. The framework, called Mul2vec, learns semantic representations of entities from their text and modality-specific features. Mul2vec works by aligning the contextual and semantic representations of entities using a neural tensor factorization (NTF) model. The NTF model learns latent factors that capture the global information of entities, which are then used to learn the semantic representations. Mul2vec is able to learn robust representations even when some of the modality-specific features are missing or noisy.

**Strengths:**

1. It proposes a novel framework for learning multi-modal representations under incomplete data. Mul2vec learns semantic representations of entities from their text and modality-specific features, even when some of the features are missing or noisy. This is a challenging problem, but Mul2vec is able to address it by aligning the contextual and semantic representations of entities using a neural tensor factorization (NTF) model.

2. Mul2vec is able to learn representations that are invariant to noise and missing data. This is because Mul2vec learns latent factors that capture the global information of entities, which are then used to learn the semantic representations. This makes Mul2vec suitable for a variety of downstream tasks, even when the training data is incomplete or noisy.

3. It is compatible with pre-trained models. Mul2vec can be used to learn semantic representations from the text and modality-specific features of entities, even if the pre-trained model was not trained on the same data. This is because Mul2vec learns the latent factors that capture the global information of entities, which are then used to learn the semantic representations. This makes Mul2vec a versatile tool that can be used to learn multi-modal representations for a variety of tasks.

**Weaknesses:**

1. The writing of the paper is below the bar, there are many typos and grammar errors, i.e., prameter -> parameter in Figure 1; many terms are written in short in the first place which is hard to understand, such as NTF, DLM and ILM in abstract.

2. Many terms and equations are not well-explained: how to use global information as topic? What are the example of topic? Others please refer to Questions section.

3. The experiments are limited in dataset and different type of downstream tasks, since the authors claim that Mul2vec learns semantic representations. Why only demonstrated the results on image-text retrieval tasks on two datasets? Also authors mentioned that their goal is to applied to multimodal tasks under incomplete data, is there any ablation studies on incomplete dataset? Or how did the authors set up experiments as noisy data or missing modality information?

4. The discussion and limitations sections showed some models, datasets and results which were not discussed in the paper.

**Questions:**

1. In section 3.3, what are the gaps between texts?
2. In section 3.4, how to use negative vector e_n, and is e_p is v_p in equation 6? I think authors need to clarify all the terms clearly in equations.
3. In Table 1, what are the differences between D and L, what are documents are ids? are they the captions in COCO and Flickr30 dataset?
4. Are there any other algebraic operations in experiments? How these algebraic operations verify the representation learning ability of Mul2vec?
5. In Table 4, how to get the topics?

---

> ### Author Response · Authors · 2023-11-12
> **Rebuttal for Reviewer BQ2D**
>
> Thank you for reviewing our paper. We'd like to respond to the comments here.
>
> ## Weaknesses:
> **1) The writing of the paper is below the bar, there are many typos and grammar errors, i.e., prameter -> parameter in Figure 1; many terms are written in short in the first place which is hard to understand, such as NTF, DLM and ILM in abstract.**
> \
> Sorry about both typos and errors. we will correct the areas pointed out or noticed.
> Due to space limitations we put the abbreviation in the abstract, but not in it.
> \
> \
> **2-1) Many terms and equations are not well-explained: how to use global information as topic?**
> \
> Mul2vec learns semantic representations through NTF, where NTF discovers topics as the global information.
> As stated in 1, *``Unlike other models, Mul2vec uses latent variables to quantify global information as topics, where these topics are discovered from context-independent information, word-occurrence, and sharing between entities.''*
> More specifically, we create *``the document-word-entity's ID tensor $\bf{\mathcal{D}} \in \mathbb{R}^{N \times V \times L}$ for $N$ documents, $V$ words, and $L$ IDs, and feed it to Mul2vec.
> Then NTF decomposes this 3rd order tensor into three latent factor matrices $\Theta=\{\theta_{n,k}\} \in \mathbb{R}^{N \times K}$, $\bf{T}=\{t_{v,k}\} \in \mathbb{R}^{V \times K}$ and $\bf{G}=\{g_{l,k}\} \in \mathbb{R}^{L \times K}$, where the $k$-th row vector of each matrix corresponds to the $k$-th factor of the tensor''*.
> \
> **2-2) What are the example of topic?**
> \
> As stated in the paper and in the previous repley, topic is a vector representation (i.e.,$\theta_{n},g_{l},t_{v}$) in our approach, like contextual representations (i.e., embeddings), and is not readable.
> \
> **2-3) Others please refer to Questions section.**
> \
> As we answered in the question section, please see this section.
> \
> \
> **3-1) The experiments are limited in dataset and different type of downstream tasks, since the authors claim that Mul2vec learns semantic representations. Why only demonstrated the results on image-text retrieval tasks on two datasets?**
> \
> For fair comparison and to ensure reproducibility and model comprehensiveness,
> this experiment dataset and settings were limited to a representative test data set and basic tasks.
> \
> **3-2) Also authors mentioned that their goal is to applied to multimodal tasks under incomplete data, is there any ablation studies on incomplete dataset? Or how did the authors set up experiments as noisy data or missing modality information?**
> \
> Yes. As shown in 4.2., we prepare the missing modality information as; *``To evaluate the effect of semantic representations,
> we also prepared the original data and the data with 20\% of each of these images and texts randomly masked, \textbf{incomplete data}, and compared the improvement obatained over the models without Mul2vec.''*
> \
> \
> **4) The discussion and limitations sections showed some models, datasets and results which were not discussed in the paper.**
> \
> Since we omitted these models, datasts, and results because they was not directly relevant.
> We will add them in the next supplementary material.
>
> # Questions:
> **1) In section 3.3, what are the gaps between texts?**
> \
> The gaps mean the differences between texts with the same content but different wording.
> \
> \
> **2) In section 3.4, how to use negative vector e_n, and is e_p is v_p in equation 6? I think authors need to clarify all the terms clearly in equations.**
> \
> Because it is common in contrastive learning and we have simplified,
> $v_{p}$ is the semantic representation of DID (traget $l$-th ID's document) and $v_{p}$ is the semantic representation of the DID in the same batch (not traget $l$-th ID's document).
> \
> **3) In Table 1, what are the differences between D and L, what are documents are ids? are they the captions in COCO and Flickr30 dataset?**
> \
> Yes. $D$, and $L$ denotes the number of captions, and ID, respectively.
> As ID is the identification of images, it is the same number as images.
> \
> \
> **4) Are there any other algebraic operations in experiments? How these algebraic operations verify the representation learning ability of Mul2vec?**
> \
> Yes. Such operations can also evaluate the model's ability, but this was not the main objective and we do not show these examples in the main results because there was no dataset available to quantitatively evaluate this abliity.
> \
> \
> **5) In Table 4, how to get the topics?**
> \
> While they are vector representations that cannot be interpreted directly like other latent variable models,
> Mul2vec learns and gets them as (i.e.,$\theta_{n},g_{l},t_{v}$), shown in Figure 1.

---

> > ### Comment · Reviewer_BQ2D · 2023-11-22
> >
> > Thanks for the detailed explanation to my questions. I've carefully read your reply and other reviewers' comments. I think there are a lot of revisions needed for this paper, like more detailed explanations of the terms and notions. Thus, I keep my evaluation ratings.

---

> > > ### Author Response · Authors · 2023-11-22
> > > **Reply to Reviewer BQ2D**
> > >
> > > Thank you for the comment
> > >
> > > If you have read the paper, other reviewers' comments, or our reply,
> > > please point out specific issues or areas that can be discussed, not just your impressions.

---

### Meta-Review · Area_Chair_tegG · 2023-12-17

**Metareview:**

This paper proposes a dual-level alignment for learning multi-modal representations under incomplete data. Considering massive weaknesses in this paper, all reviewers give reject comments. The AC thus decided to reject it.

**Justification For Why Not Higher Score:**

The writing, experiments and novelty are far below the bar.

**Justification For Why Not Lower Score:**

N/A

---

### Decision · Program_Chairs · 2024-01-16

Reject